# Modulation of electrical activity of proteinoid microspheres with chondroitin sulfate clusters

**Panagiotis Mougkogiannis**[ORCID][*], **Andrew Adamatzky**

Unconventional Computing Lab, University of the West of England, Bristol, United Kingdom

These authors contributed equally to this work.
* Panagiotis.Mougkogiannis@uwe.ac.uk

**Data Availability Statement:** The data for the paper is available online 452 and can be accessed at https://zenodo.org/records/12805409.

**Funding:** The research was supported by EPSRC Grant EP/W010887/1 "Computing with 512

## Abstract

Proteinoids—thermal proteins—are produced by heating amino acids to their melting point and initiation of polymerisation to produce polymeric chains. Proteinoids swell in aqueous solution into hollow microspheres. The proteinoid microspheres produce endogenous burst of electrical potential spikes and change patterns of their electrical activity in response to illumination. These microspheres were proposed as proto-neurons in 1950s. To evaluate pathways of potential evolution of these proto-neurons and their applicability of chimera neuromorphic circuits we decided to hybridise them with hondroitin sulphate (CS) clusters, which form a part of the brain extracellular matrix. We found a novel synergistic interaction between CS clusters and proteinoids that dramatically affects patterns of electrical activity of proteinoid microspheres. Our study might shed light on evolution of synaptic plasticity's molecular mechanisms and the role of extracellular matrix-protein interactions in learning, and open up possibilities for novel methods in unconventional computing and the development of adaptable, brain-inspired computational systems.

## Introduction

The complex mechanisms underlying synaptic plasticity and learning [1–4] continue to be a central focus in neuroscience research. Interesting recent study has uncovered fascinating insights into the complex connection between components of the extracellular matrix and synaptic proteins [5–7]. This discovery has revealed a dynamic environment that plays a crucial role in shaping neuronal communication and plasticity [8]. Two important factors in synaptic function are chondroitin sulphate proteoglycans (CSPGs) and proteinoids. However, their role in learning mechanisms is not well understood [9, 10].

Chondroitin sulfate, a sulfated glycosaminoglycan, plays a crucial role in the neural extracellular matrix, influencing neuronal development and plasticity [11]. The CS clusters, with their varied sulfation patterns, form a diverse microenvironment that impacts receptor mobility, ion diffusion, and growth factor sequestration [12]. Recent work has shown that changes in the composition of CS can have a significant impact on synaptic plasticity and learning behaviours. This suggests that these molecules play a crucial role in the processing and storage of information [13].

proteinoids". The funders had no role in study design, data collection and analysis, 513 decision to publish, or preparation of the manuscript.

**Competing interests:** The authors have declared that no competing interests exist.

Proteinoids are a type of thermal proteins that have attracted interest due to their possible involvement in the beginning of life and, more recently, their significance in brain function [14]. The proteins synthesised abiotically show remarkable similarities to biological proteins in both their structure and function [15]. Recent findings indicate that proteinoids have the ability to interact with neuronal membranes and impact synaptic transmission. This suggests that proteinoids could possibly act as basic building blocks for complex synaptic proteins [16].

The proteinoid microspheres maintain a steady state membrane potential 20 mV to 70 mV without any stimulating current and some microspheres in the population display the opposite polarization steadily [17]. Electrical membrane potentials, oscillations, and action potentials are observed in the microspheres impaled with microelectrodes. These microspheres exhibit action-potential like spikes. The electrical activity of the microspheres also includes spontaneous bursts of electrical potential (flip-flops), and miniature potential activities at flopped phases [18]. Although effects are of greater magnitude when the microspheres contain glycerol lecithin, purely synthetic microspheres can attain 20 mV membrane potential [19]. The neuron-like spiking behaviour of proteinoid microspheres led researchers to consider them as proto-neurons [14, 20–22].

An extracellular matrix of CS clusters plays role as an active contributor to synaptic activity, rather than just a passive support structure [7]. Therefore, it is important to study how CS clusters might affect information processing, expressed in the form of electrical spiking patterns, in ensembles of proteinoid microspheres. The results of our research may offer new and unique understandings of the molecular mechanisms involved in learning and memory formation [23, 24]. This might potentially lead to the development of new approaches for treating neurological illnesses that are linked to problems with synapse function.

Prior research showed that proteinoid microspheres can produce electrical spikes [25–28]. Our study greatly enhances this knowledge using various novel methods. We use advanced experimental methods, like high-precision electrochemical techniques and multi-parametric imaging. They provide a detailed view of the electrical activity in these microspheres. Secondly, we present a new framework. It integrates molecular mechanics optimization, solvation analysis, and modified Hodgkin-Huxley models. This method models the complex interactions at a molecular level. It yields new insights into the mechanisms that govern the electrical properties. Our research is the first to study how chondroitin sulphate concentration affects the electrical properties of proteinoid microspheres. It uncovers a new aspect of their function. We use the Izhikevich neuron model in a new way. It mimics thalamocortical dynamics in these systems. It connects proteinoid behaviour and neuronal signaling. These developments may reveal new ways to process signals and create "memories" in proto-cells. They could change our understanding of the origins of neural-like behaviour. This may lead to bio-inspired computer systems.

Reservoir computing with spiking neural networks shows promise. It can model chaotic dynamics and forecast time series. Reservoir computing systems can analyze complex, time-dependent data. This is due to the stochastic network architecture and intrinsic temporal dynamics of spiking neurons [29–31]. Our proteinoid microsphere networks have spontaneous, regulated electrical activity. They may have similar computing powers. The random arrangement of microspheres and the unstable links from chondroitin sulfate clusters may act as a physical reservoir. The spiking activity of individual microspheres modifies incoming signals in a nonlinear way. This study focuses on the electrical properties of proteinoid microsphere networks. Future research may explore their use as novel computing devices. This may mean studying how different network architectures affect processing. Also, it may include looking at the effects of external modulations. Our proteinoid system's computation may

outperform neuron models in reservoir computing. This could reveal insights into both biological and artificial information processing.

# Materials and methods

## Preparation of chondroitin sulfate-proteinoid mixture

The chondroitin sulfate-proteinoid mixture was synthesized by preparing individual solutions of chondroitin sulfate and proteinoid, and then mixing the solutions to make the final formulation. Four chondroitin sulfate-proteinoid samples were produced by dissolving various amounts of chondroitin sulfate (11 mg, 18.6 mg, 40 mg, and 102 mg) in 5 ml of dimethyl sulfoxide (DMSO) from Sigma Aldrich (CAS Number: 67-68-5, EC Number: 200-664-3, Molecular Weight: 78.13 gr/mol). An analytical balance was used to weigh the chondroitin sulfate (CAS Number: 9007-28-7, Molecular Weight: 10,000-50,000 gr/mol), which was then transferred to clean and dry beakers. The mixtures were subsequently stirred using a magnetic stirrer at ambient temperature until the chondroitin sulfate was fully dissolved. A 5 ml proteinoid solution containing L-Glutamic Acid (L-Glu), L-Phenylalanine (L-Phe), and L-Aspartic Acid (L-Asp) has been made in a separate beaker. The proteinoid was well dissolved in the water solution. The chondroitin sulfate solutions, which were made in DMSO, were introduced gradually to the beaker that contained the proteinoid solution in water. The solutions were thoroughly mixed by gently stirring them with a magnetic stirrer. The stirring process was prolonged for a duration of 5-10 minutes in order to ensure thorough and complete mixing of the chondroitin sulfate-proteinoid blends. The synthesized chondroitin sulfate-proteinoid mixtures were then ready for further characterization and measurements.

## Proteinoid-chondroitin sulfate solution concentrations

Four proteinoid-chondroitin sulfate solutions were prepared with varying concentrations of chondroitin sulfate. The total volume of each solution was 10 ml, consisting of 5 ml DMSO and 5 ml proteinoid solution. The concentrations of chondroitin sulfate in the solutions were 1.1 mg/ml (1100 $\mu$g/ml), 1.86 mg/ml (1860 $\mu$g/ml), 4 mg/ml (4000 $\mu$g/ml), and 10.2 mg/ml (10200 $\mu$g/ml), corresponding to 11 mg, 18.6 mg, 40 mg, and 102 mg of chondroitin sulfate in 10 ml total volume, respectively.

## Electrochemical characterization setup

The electrochemical characterization setup, as shown in Fig 1, was employed to measure the voltage responses of the proteinoid-chondroitin sulfate solutions at the aforementioned concentrations. The setup consisted of a container holding the proteinoid-chondroitin sulfate solution, with two needle electrodes (Pt and Ir coated stainless steel wires) placed 10 mm apart. A high-precision 24-bit ADC data recorder was used to record the voltage responses from the electrodes. A heating block was incorporated into the setup to control and monitor the temperature of the container. This allowed for the simultaneous registration of thermal and electrical parameters during the characterization process. The ADC data recorder's high sensitivity enabled the detection of tiny voltage variations in the $\mu$V range, facilitating the mapping of spatiotemporal voltage responses in the proteinoid-chondroitin sulfate system at different concentrations of chondroitin sulfate.

## Voltage response measurements

The voltage responses of the proteinoid-chondroitin sulfate solutions were measured using the electrochemical characterization setup described in Fig 1. The measurements were

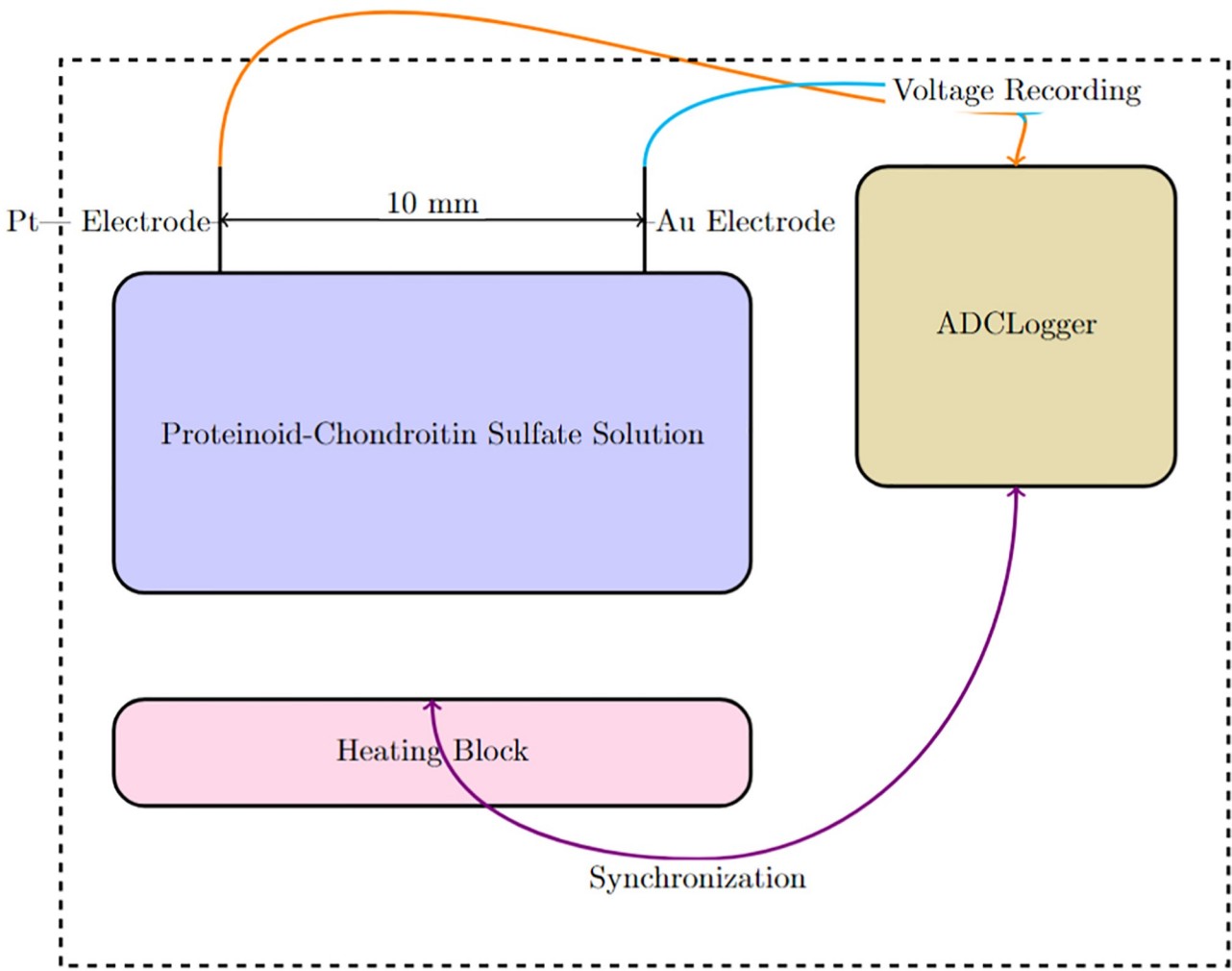

**Fig 1. Proteinoid-chondroitin sulfate electrochemical characterization setup schematic.** A container holds the proteinoid-chondroitin sulfate solution and two needle electrodes (Pt and Ir coated stainless steel wires) 10 mm apart. A high-precision 24-bit ADC data recorder records voltage responses from the electrodes. A heating block controls and monitors container temperature. The heating block and data logger register thermal and electrical parameters simultaneously. By detecting tiny voltage variations in the $\mu$V range, this technique can map spatiotemporal voltage responses in the proteinoid-chondroitin sulfate system.

conducted at room temperature (approximately 25°C) for each of the four concentrations of chondroitin sulfate. The ADC data recorder collected voltage data over a period of 20 days for each solution, with a sampling rate of 1 Hz. The recorded data was then processed and analyzed to identify any distinct patterns or trends in the voltage responses corresponding to the different concentrations of chondroitin sulfate in the proteinoid-chondroitin sulfate system.

## Results

### Molecular mechanics optimization and solvation analysis of chondroitin sulfate-proteinoid interactions

The molecular mechanics optimization using the MMFF94$_{aq}$ force field provided valuable insights into the energetics and structural properties of CS, the proteinoid (L-Glu:L-Asp:

**Table 1. Molecular mechanics optimization results for chondroitin sulfate, proteinoid, and their complex.**

| System | Final Energy (kJ/mol) | Solvation Energy (kJ/mol) | Dipole Moment (Debye) |
|---|---|---|---|
| Chondroitin Sulfate | −58.02 | −183.86 | 4.190 |
| Proteinoid | −6.21 | −163.01 | 9.212 |
| CS-Proteinoid Complex | −137.18 | −326.25 | 18.880 |

L-Phe), and their complex. Table 1 summarizes the key results of these simulations. Chondroitin sulfate, with a stoichiometry of $C_{13}H_{21}NO_{15}S$, achieved an optimized structure after 79 cycles. The final energy of −58.02 kJ/mol indicates a stable conformation. Notably, the solvation energy of −183.86 kJ/mol suggests strong interactions with the aqueous environment, which is expected for this highly polar molecule. The dipole moment of 4.190 Debye reflects its charge distribution and potential for electrostatic interactions. The proteinoid, composed of L-Glu, L-Asp, and L-Phe, required 133 optimization cycles to reach its final conformation. Its final energy of −6.21 kJ/mol is higher than that of CS, indicating a less stable structure in isolation. The solvation energy of −163.01 kJ/mol, while significant, is lower than that of CS, suggesting less intense interactions with the aqueous environment. The higher dipole moment of 9.212 Debye indicates a more pronounced charge separation within the molecule. The CS-proteinoid complex, with a stoichiometry of $C_{33}H_{50}N_4O_{22}S$, required substantially more optimization cycles (1009) to reach its final state. This increased computational effort suggests a more complex energy landscape and potential conformational changes during the interaction. The final energy of −137.18 kJ/mol is significantly lower than the sum of the individual components (−64.23 kJ/mol), indicating a favorable interaction between CS and the proteinoid. Most striking is the solvation energy of the complex (−326.25 kJ/mol), which is substantially more negative than the sum of the individual solvation energies (−346.87 kJ/mol). This suggests that the formation of the complex slightly reduces the overall interaction with the aqueous environment, possibly due to the burial of some polar groups at the interface between CS and the proteinoid. The dipole moment of the complex (18.880 Debye) is notably higher than the sum of the individual components (13.402 Debye). This increase indicates a significant redistribution of charge upon complex formation, which could play a crucial role in its interactions with other biomolecules and its potential function in synaptic processes.

The molecular mechanics simulations demonstrate a complex interaction between chondroitin sulphate and the proteinoid. The substantial reduction in overall energy during the formation of the complex indicates a high affinity between these molecules, which may play a critical role in their co-localization and function in synaptic environments. The reorganisation of electric charge, as indicated by the heightened dipole moment, could enhance interactions between proteinoid microspheres. Moreover, the modified solvation energy of the complex suggests a rearrangement of water molecules at the interface between the CS-proteinoid. This reorganisation has the ability to generate separate microenvironments within synapse-like structures, which could have an impact on the local concentrations of ions or small molecules. Changes in the local physiochemical environment can potentially influence synaptic plasticity mechanisms (Fig 2).

## Impact of chondroitin sulfate dosage on proteinoid spiking dynamics

The effect of chondroitin sulfate concentration on the raw potential of proteinoid solutions was investigated, and the results are presented in Fig 3. The statistical analysis of the measured potentials for each concentration reveals that the 1.86 mg/ml solution exhibits the highest

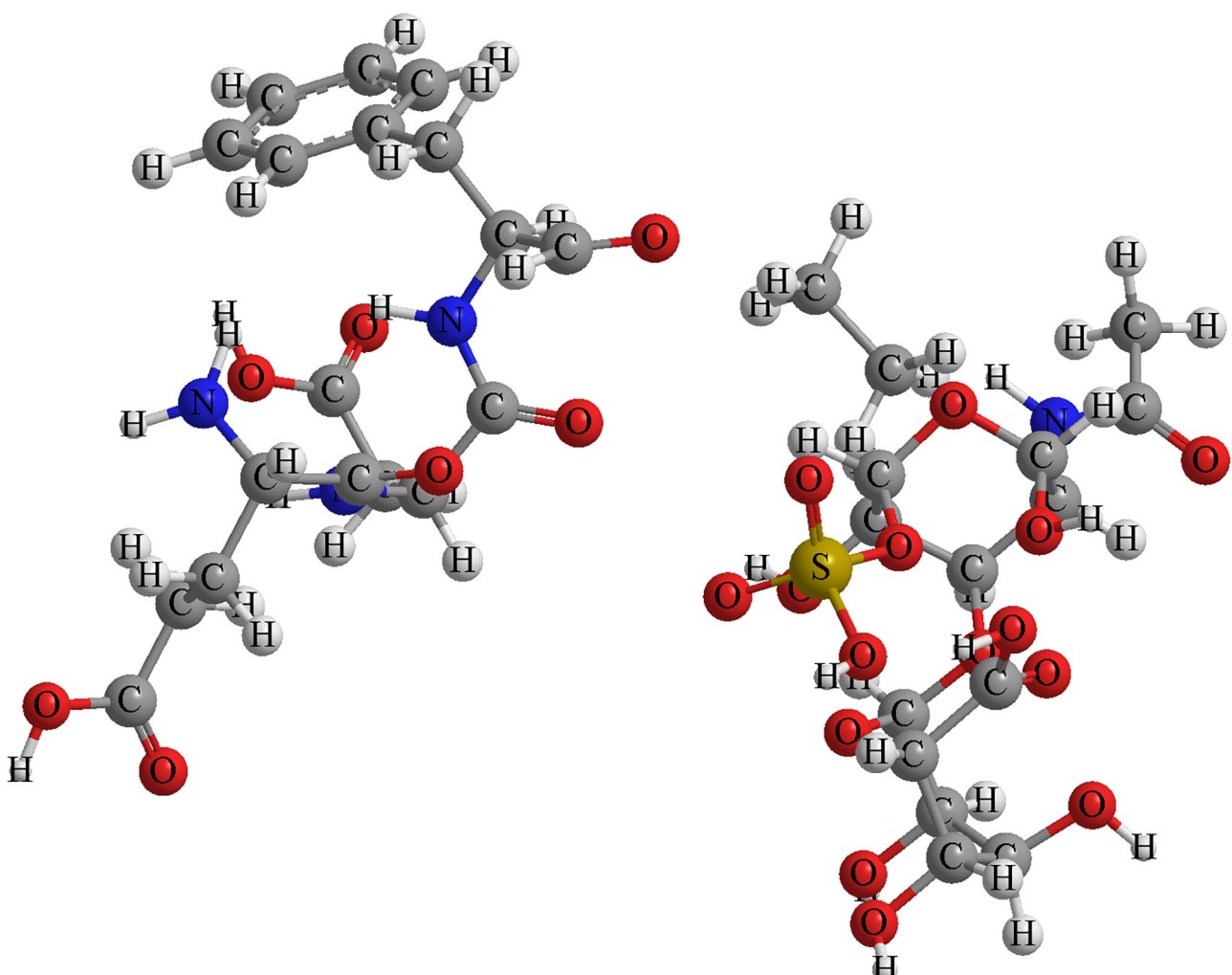

**Fig 2. Molecular structures of proteinoid and chondroitin sulfate components.** Molecular structures of the proteinoid L-Glu:L-Asp:L-Phe (left) and chondroitin sulfate (right). The proteinoid structure shows the peptide backbone with side chains of glutamic acid, aspartic acid, and phenylalanine. The chondroitin sulfate structure illustrates its characteristic sugar backbone with sulfate groups. These molecules interact to form complexes that may influence synaptic plasticity and learning mechanisms.

variation in potential, with a standard deviation of 4.52946 V and a range from −14.445 V to 8.405 V. This suggests that the 1.86 mg/ml concentration of chondroitin sulfate has the most significant impact on the electrical properties of the proteinoid solution. In contrast, the 10.20 mg/ml solution displays the least variation in potential, with a standard deviation of 0.1075 V and a range from −0.503 V to 0.099 V. This indicates that the highest concentration of chondroitin sulfate tested (10.20 mg/ml) results in a more stable and consistent potential compared to lower concentrations and the water control. The 1.10 mg/ml and 4.00 mg/ml solutions show intermediate levels of variation in potential, with standard deviations of 1.20925 V and 1.00696 V, respectively. The water control exhibits the least variation among all samples, with a standard deviation of 0.02097 V, serving as a baseline for comparing the effects of chondroitin sulfate on the proteinoid solution's electrical properties.

Chondroitin sulphate is a sulfated glycosaminoglycan with a complex structure and various biological activities [32]. The negatively charged sulphate groups on the chondroitin sulphate

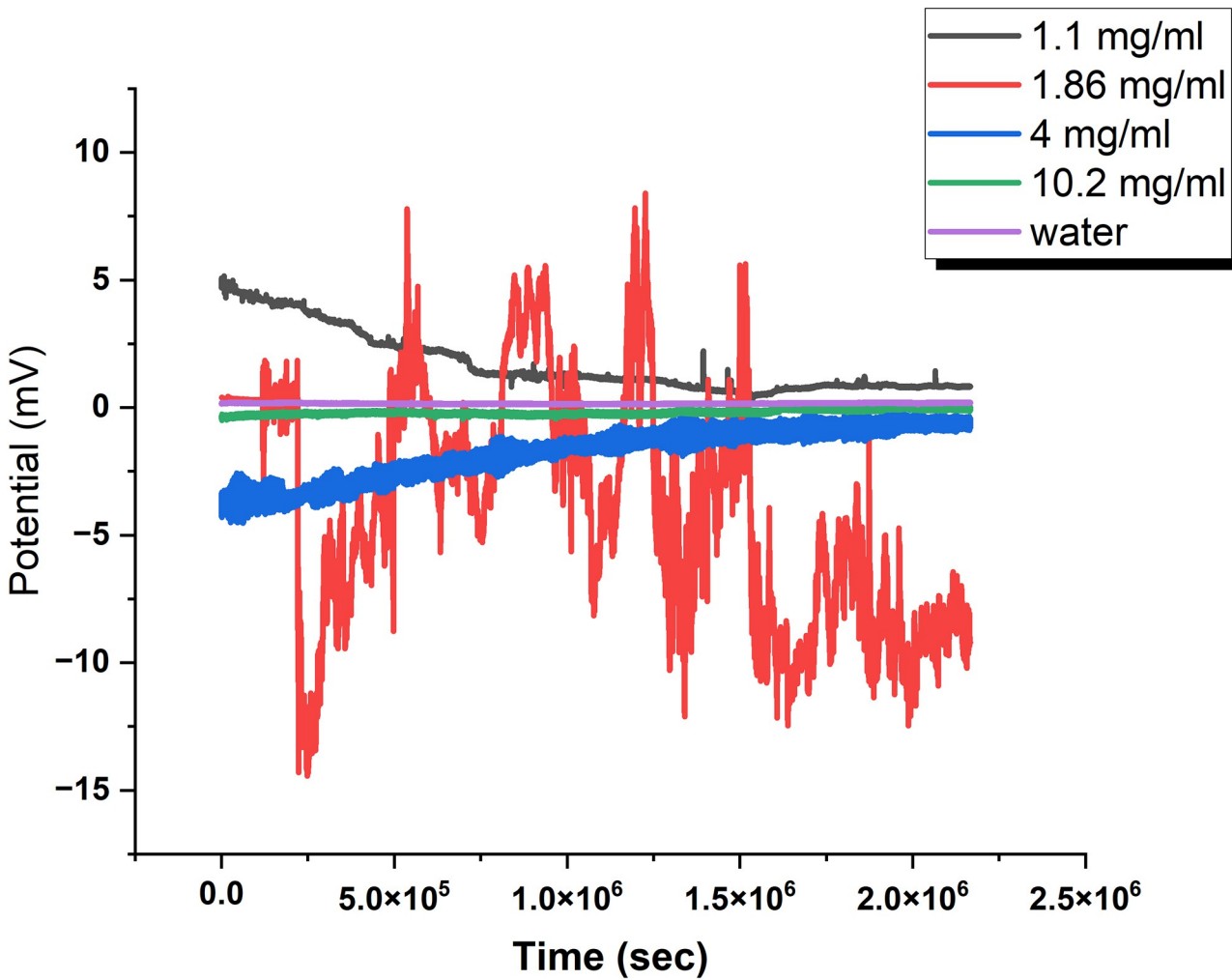

**Fig 3. Effect of chondroitin sulfate concentration on the raw potential of proteinoid solutions.** This figure demonstrates the effect of chondroitin sulfate concentration on the raw potential of proteinoid solutions. The statistical analysis of the measured potentials for each concentration reveals that the 1.10 mg/ml solution has a mean potential of 1.70424 V, a standard deviation of 1.20925 V, and a range from 0.346 V to 5.164 V. The 1.86 mg/ml solution exhibits a mean potential of −3.91245 V, a standard deviation of 4.52946 V, and a range from −14.445 mV to 8.405 mV. The 4.00 mg/ml solution shows a mean potential of −1.74356 V, a standard deviation of 1.00696 V, and a range from −4.537 V to −0.204 V. The 10.20 mg/ml solution has a mean potential of −0.18757 V, a standard deviation of 0.1075 V, and a range from −0.503 V to 0.099 V. The water control displays a mean potential of 0.15751 V, a standard deviation of 0.02097 V, and a range from 0.107 V to 0.217 V. These results indicate that the addition of chondroitin sulfate significantly alters the electrical properties of the proteinoid solution, with the highest concentration (10.20 mg/ml) exhibiting the least variation in the measured potential compared to lower concentrations and the water control.

chain can interact with the charged amino acid residues of the proteinoid molecules, resulting in the formation of electrostatic complexes [33]. These interactions have the potential to modify the form and structure of the proteinoid molecules, leading to variations in their electrical characteristics. The variability in the inherent potential of the proteinoid solutions is most evident when the concentration of chondroitin sulphate is 1.86 mg/ml. This indicates that this concentration range may be a crucial threshold for the formation of these electrostatic complexes. At lower concentrations, such as 1.10 mg/ml, the interactions between chondroitin sulphate and proteinoid molecules may not be strong enough to cause notable alterations in the electrical characteristics of the solution. On the other hand, when the concentration of chondroitin sulphate is higher, like 4.00 mg/ml and 10.20 mg/ml, it can fill almost all the binding

sites on the proteinoid molecules. This results in a more stable and consistent potential. The little fluctuation in electrical potential observed at the highest concentration of chondroitin sulphate (10.20 mg/ml) could be explained by the formation of a durable and uniform mixture between the chondroitin sulphate and proteinoid molecules. This complex may display a homogeneous distribution of charges, leading to a more uniform electrical potential throughout the solution. In addition, the abundance of chondroitin sulphate in the solution may enhance the overall ionic strength, hence providing further stability to the electrical characteristics of the proteinoid-chondroitin sulphate complex [34]. The water control is used as a reference point to assess the impact of chondroitin sulphate on the electrical characteristics of the proteinoid solution. The negligible fluctuation in electrical potential seen in the water control indicates that the water molecules alone do not have a substantial impact on the instability of the solution's electrical properties in the absence of chondroitin sulphate.

Fig 4 shows a non-linear correlation. The mean electrical potential is related to CS concentration. The minimum potential was −3.91 V at 1.86 mg/ml. The maximum was 1.70 V at 1.10 mg/ml. The standard deviation of electrical potential significantly diminishes with elevated CS concentration. This shows that high CS concentrations stabilize the electrical properties of the proteinoid solutions. The water control (0 mg/ml CS) exhibited the minimal fluctuation, establishing a baseline for comparison.

**Characterization of chondroitin sulfate—Proteinoid interaction at 1.1 mg/ml concentration.** The analysis of spike data used a thorough statistical method. It found the amplitude

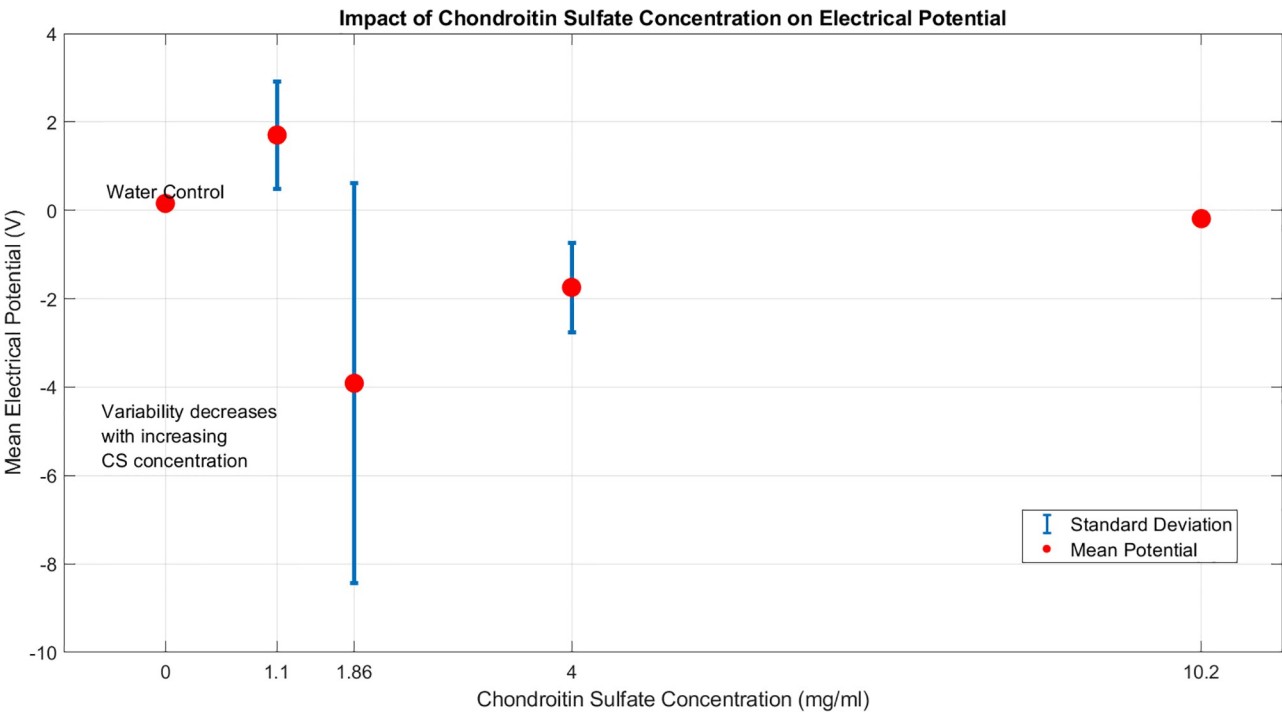

**Fig 4. Concentration-dependent effects of chondroitin sulfate on electrical potential in proteinoid solutions.** The graph shows the link between chondroitin sulphate (CS) concentration and the average electrical potential of proteinoid solution. Data points denote mean potentials (red circles) and standard deviations (error bars) for each tested CS concentration (1.10, 1.86, 4.00, and 10.20 mg/ml), plus a water control (0 mg/ml). The plot shows a non-linear correlation between CS concentration and electrical potential. It is most variable at moderate concentrations (1.86 mg/ml). The electrical potential fluctuates less with higher CS concentrations. This suggests a stabilising effect at high concentrations. The water control demonstrates minimal variability, serving as a baseline for comparison. This visualisation shows a complex link between CS concentration and electrical traits in proteinoid solutions. It highlights CS's potential role in regulating and stabilising electrical activity in these systems.

and period distributions of the recorded electrical activity. For each distribution, we computed key stats: mean, median, standard deviation, and quartiles. This showed the data's central tendency and spread. To clarify the distributions' traits, we calculated their skewness and kurtosis. Skewness measures a distribution's asymmetry. We computed it using the given equation:

$$\text{Skewness} = \frac{\mathbb{E}[(X - \mu)^3]}{\sigma^3} \tag{1}$$

where $X$ is the random variable, $\mu$ is the mean, $\sigma$ is the standard deviation, and $\mathbb{E}$ denotes the expected value. Kurtosis, which quantifies the tailedness of the distribution, was computed using:

$$\text{Kurtosis} = \frac{\mathbb{E}[(X - \mu)^4]}{\sigma^4} \tag{2}$$

These metrics provide key insights into the amplitude and period distributions. They help us better interpret the spiking behaviour. We applied a gamma distribution to the amplitude data. This characterized its probability. We used maximum likelihood estimation to find the shape and rate parameters.

Fig 5 presents the spiking behaviour and statistical analysis of the proteinoid-chondroitin sulfate system. In Fig 5a, the amplitude of the spikes (in mV) is plotted against time (in seconds), revealing the temporal distribution of the spike events. The plot showcases the stochastic nature of the spiking behaviour, with spikes occurring at irregular intervals and varying amplitudes. The statistical analysis of the spike amplitudes is depicted in Fig 5b, which displays the gamma distribution of the amplitudes with shape parameter $a = 32.27402$ and rate parameter $b = 0.01904$. The gamma distribution provides an excellent fit to the observed amplitude data, indicating that the spike amplitudes follow a specific probability distribution. The

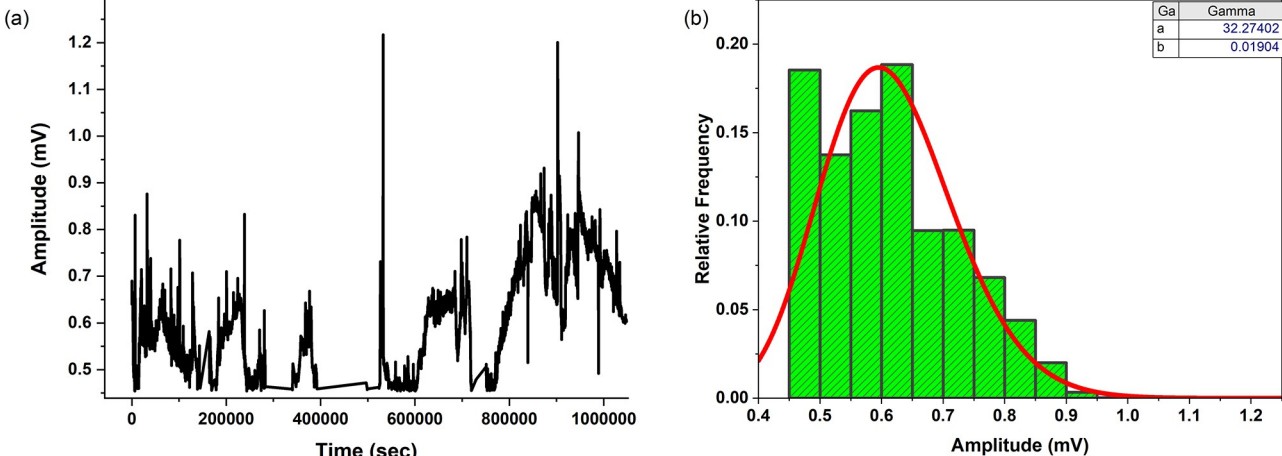

**Fig 5. Effects of chondroitin sulfate (1.1 mg/ml) on the electrical potential of proteinoid solutions.** Fig 5 displays the spiking behaviour and statistical analysis of the proteinoid-chondroitin sulphate system. The graph in Fig 5a displays the relationship between the magnitude of the spikes (measured in millivolts) and the corresponding time (measured in seconds), providing insight into the temporal pattern of the spiking episodes. The graphic demonstrates the chaotic character of the spiking behaviour, with spikes happening at irregular intervals and with different magnitudes. The amplitude data has quartiles of 0.52 mV, 0.60 mV, and 0.69 mV. The mean value is 0.61 mV, the maximum is 1.22 mV, and the minimum is 0.45 mV. The amplitude's standard deviation is 0.11 mV, indicating a relatively small range of values around the average. Moreover, the statistical examination of the spiking periods, which refers to the time interval between consecutive spikes, demonstrates a broad spectrum of values. The quartiles of the period data are 175.00 seconds, 193.00 seconds, and 249.00 seconds, and the mean value is 291.80 seconds. Significantly, the highest recorded duration is 104762.00 seconds, suggesting the existence of exceptionally lengthy gaps between specific episodes of spiking. The minimum duration is 92.00 seconds, while the standard deviation of the durations is 2091.22 seconds, indicating a significant variation in the intervals between spikes.

quartiles of the amplitude data are 0.52 mV, 0.60 mV, and 0.69 mV, with a mean value of 0.61 mV, a maximum of 1.22 mV, and a minimum of 0.45 mV. The standard deviation of the amplitudes is 0.11 mV, suggesting a relatively narrow spread around the mean. Furthermore, the statistical analysis of the spiking periods (time between consecutive spikes) reveals a wide range of values. The quartiles of the period data are 175.00 seconds, 193.00 seconds, and 249.00 seconds, with a mean value of 291.80 seconds. Notably, the maximum period observed is 104762.00 seconds, indicating the presence of extremely long intervals between some spiking events. The minimum period is 92.00 seconds, and the standard deviation of the periods is 2091.22 seconds, reflecting the high variability in the spiking intervals.

The skewness and kurtosis values were calculated for both the amplitude and period distributions. The amplitude data exhibited a skewness of 0.5921, indicating a moderately right-skewed distribution. This suggests the presence of some higher amplitude values in the data, with the tail of the distribution extending towards the right. The kurtosis of the amplitude data was found to be 2.9488, which is close to the kurtosis of a standard normal distribution. This implies that the peakedness of the amplitude distribution is similar to that of a normal distribution. In contrast, the period data showed a remarkably high skewness of 41.1538, revealing a strong asymmetry in the distribution. The highly positive skewness indicates a long tail on the right side of the distribution, suggesting the presence of extremely large period values compared to the majority of the data. Furthermore, the kurtosis of the period data was calculated to be 1892.4993, which is significantly higher than the kurtosis of a normal distribution. This extremely high kurtosis value indicates a highly leptokurtic distribution, characterized by a sharp peak and heavy tails. The presence of outliers or extreme values in the period data is strongly supported by these statistical measures. The gamma distribution, which is commonly used to model skewed and non-negative data, can be employed to describe the amplitude and period distributions. The probability density function of the gamma distribution is given by:

$$f(x; \alpha, \beta) = \frac{\beta^{\alpha}}{\Gamma(\alpha)} x^{\alpha-1} e^{-\beta x}, \quad x > 0 \tag{3}$$

where $\alpha > 0$ is the shape parameter, $\beta > 0$ is the rate parameter, and $\Gamma(\alpha)$ is the gamma function, defined as:

$$\Gamma(\alpha) = \int_0^{\infty} x^{\alpha-1} e^{-x} dx \tag{4}$$

The statistical analysis of the spike amplitudes is shown in Fig 5b. The figure illustrates the gamma distribution of the amplitudes, with a shape parameter of $a = 32.27402$ and a rate parameter of $b = 0.01904$. The gamma distribution is a highly suitable model for the observed amplitude data, suggesting that the spike amplitudes follow to an unique probability distribution. The moderate right-skewness and normal-like peakedness of the amplitude distribution suggest that the system exhibits a range of amplitude values, with some higher amplitudes occurring more frequently than in a purely random process. On the other hand, the strong right-skewness and high kurtosis of the period distribution indicate the presence of rare but extremely long periods between electrical potential spikes, which may be indicative of complex underlying dynamics or long-range correlations within the system. Further investigation into the physical and chemical mechanisms governing these statistical properties could provide valuable insights into the self-organization and information processing capabilities of the chondroitin sulfate-proteinoid system, and may have implications for the development of novel biomaterials and computing paradigms based on such systems.

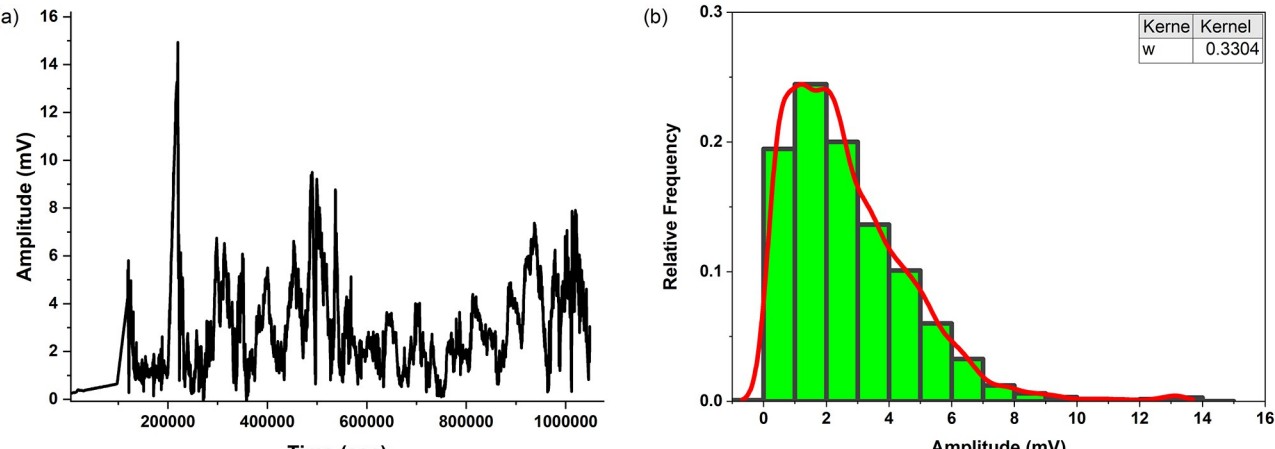

**Fig 6. Effects of chondroitin sulfate (1.8 mg/ml) on the electrical potential of proteinoid solutions.** a) This graph displays an analysis of the spiking activity in the proteinoid-chondroitin sulfate system. Fig 6a depicts the magnitudes of the spikes (measured in millivolts) as they emerge over a period of time (measured in seconds), providing insight into the frequency and range of the spiking events. The statistical examination of the spike amplitudes indicates that the mean amplitude is 5.07 mV, with quartiles at 2.86 mV, 4.96 mV, and 6.69 mV. The highest recorded amplitude is 15.01 millivolts, while the lowest is 0.38 millivolts, suggesting a broad range of spike magnitudes. The amplitude values have a standard deviation of 3.38 mV, indicating a significant degree of variability from the mean. Additionally, the peak-to-peak distance is 14.63 mV, and the root mean square (RMS) value is 6.05 mV. b) This figure displays the histogram of the spike amplitudes, offering insights into the underlying distribution of the amplitude values. A kernel smooth curve (red line) is applied to the histogram using a bandwidth of w = 0.3304. The histogram's shape and the fitted curve indicate a non-uniform distribution of spike amplitudes, with a distinct peak around the average value and a tail that skews towards larger amplitudes. The statistical analysis of the spiking periods indicates that the median period between successive spikes is 187.00 seconds, with the lower quartile at 172.00 seconds and the upper quartile at 230.50 seconds. The average inter-spike interval is roughly 3.4 minutes, which corresponds to a mean period of 203.39 seconds. The longest measured period is 308.00 seconds, while the shortest is 72.00 seconds, indicating the existence of both extended and short gaps between spiking episodes. The periods have a standard deviation of 52.22 seconds, indicating a significant level of fluctuation in the timing of spikes.

**Characterization of chondroitin sulfate—Proteinoid interaction at 1.8 mg/ml concentration.** Figs 5 and 6 present the analysis of the spiking behaviour in the proteinoid-chondroitin sulfate system with chondroitin sulfate concentrations of 1.1 mg/ml and 1.8 mg/ml, respectively. The temporal distribution of spike amplitudes, shown in Figs 5a and 6a, reveals the variability and characteristics of the spiking events over time for each concentration. For the 1.1 mg/ml concentration, the statistical analysis of the spike amplitudes indicates a mean amplitude of 0.61 mV, with quartiles at 0.52 mV, 0.60 mV, and 0.69 mV. The observed amplitudes range from a minimum of 0.45 mV to a maximum of 1.22 mV, with a standard deviation of 0.11 mV. In contrast, the 1.8 mg/ml concentration exhibits a higher mean amplitude of 5.07 mV, with quartiles at 2.86 mV, 4.96 mV, and 6.69 mV. The amplitude range for the 1.8 mg/ml concentration spans from 0.38 mV to 15.01 mV, with a standard deviation of 3.38 mV. Additionally, for the 1.8 mg/ml concentration, the peak-to-peak distance is 14.63 mV, and the root mean square (RMS) value is 6.05 mV. The histograms of spike amplitudes for both concentrations (Figs 5b and 6b) provide insights into the underlying distribution of amplitude values. The histograms are fitted with kernel smooth curves [35] using a bandwidth of w = 0.67148 for the 1.8 mg/ml concentration, while the bandwidth for the 1.1 mg/ml concentration remains unchanged at w = 0.67148.

The kernel density estimate $\hat{f}_h(x)$ at a point $x$, given a sample $x_1, x_2, \ldots, x_n$ and a bandwidth $h$, is defined as:

$$\hat{f}h(x) = \frac{1}{nh} \sum i = 1^n K\left(\frac{x - x_i}{h}\right) \tag{5}$$

where $K$ is the kernel function, and the Scott bandwidth $h$ is given by:

$$h = n^{-1/5} \cdot 1.06 \cdot \min(\hat{\sigma}, IQR/1.34) \tag{6}$$

Here, $n$ is the sample size, $\hat{\sigma}$ is the sample standard deviation, and $IQR$ is the interquartile range.

The shape of the histograms and the fitted curves suggest non-uniform distributions of spike amplitudes for both concentrations, with prominent peaks around the mean values and right-skewed tails extending towards higher amplitudes. The analysis of spiking periods reveals notable differences between the two concentrations. For the 1.1 mg/ml concentration, the median period between consecutive spikes is 193.00 seconds, with quartiles at 175.00 seconds and 249.00 seconds. The mean period is 291.80 seconds, with a standard deviation of 2091.22 seconds. Remarkably, the maximum period observed for this concentration is 104762.00 seconds, indicating the presence of exceptionally long intervals between spiking events. In comparison, the 1.8 mg/ml concentration exhibits a lower median period of 187.00 seconds, with quartiles at 172.00 seconds and 230.50 seconds. The mean period for the 1.8 mg/ml concentration is 203.39 seconds, with a standard deviation of 52.22 seconds. The maximum period observed for this concentration is 308.00 seconds, significantly shorter than that of the 1.1 mg/ml concentration.

The comparative analysis of the spiking behaviour in proteinoid-chondroitin sulfate systems with different concentrations of chondroitin sulfate highlights the concentration-dependent effects on the electrical properties and temporal dynamics of the system. The 1.8 mg/ml concentration exhibits higher spike amplitudes compared to the 1.1 mg/ml concentration, with a mean amplitude of 5.07 mV and a wider range of amplitude values spanning from 0.38 mV to 15.01 mV. The 1.8 mg/ml concentration also demonstrates a peak-to-peak distance of 14.63 mV and an RMS value of 6.05 mV, further characterizing the amplitude distribution. In terms of spiking periods, the 1.8 mg/ml concentration exhibits a lower median period of 187.00 seconds compared to the 1.1 mg/ml concentration, with quartiles at 172.00 seconds and 230.50 seconds. The mean period for the 1.8 mg/ml concentration is 203.39 seconds, with a standard deviation of 52.22 seconds. The maximum period observed for the 1.8 mg/ml concentration is 308.00 seconds, significantly shorter than the exceptionally long maximum period of 104762.00 seconds observed for the 1.1 mg/ml concentration.

These findings suggest that the concentration of chondroitin sulfate plays a crucial role in modulating the spiking activity of proteinoid-based systems. The observed differences in spike amplitudes and periods between the two concentrations indicate that higher concentrations of chondroitin sulfate may facilitate more prominent spiking behaviour, with higher amplitudes and more consistent timing of spikes. However, the exceptionally long maximum period observed for the 1.1 mg/ml concentration suggests the presence of complex dynamics and potential long-range correlations within the system at lower concentrations. These insights contribute to our understanding of the concentration-dependent effects of chondroitin sulfate on the electrical properties and temporal dynamics of proteinoid-based systems. The higher amplitudes and more consistent spiking periods observed at the 1.8 mg/ml concentration provide a basis for further investigations into the mechanisms underlying the enhanced spiking behaviour at higher chondroitin sulfate concentrations. Additionally, the presence of exceptionally long spiking periods at the 1.1 mg/ml concentration needs further investigation of the complex dynamics and potential long-range correlations within the system. Overall, these findings highlight the importance of considering the concentration of chondroitin sulfate when studying and designing proteinoid-based systems for unconventional computing and bioinspired technologies.

The skewness and kurtosis values of the amplitude and period data of the 1.8 mg/ml chondroitin sulfate proteinoid sample offer more understanding of the distribution characteristics of the spiking behaviour. The amplitude data displays a skewness value of 0.9697 and a kurtosis value of 4.3752. The positive skewness shows a moderate rightward skew in the distribution of spike amplitudes, with a longer tail on the right side towards higher amplitude values. This implies that there is a greater number of spikes with amplitudes above the average compared to spikes with amplitudes below the average. The kurtosis value of 4.3752, which is greater than 3 (the kurtosis of a regular normal distribution), suggests that the amplitude distribution is leptokurtic. This means that it has a more pronounced peak and heavier tails compared to a normal distribution. The elevated kurtosis value indicates a greater clustering of amplitude values around the mean, accompanied by a small number of extreme values in the distribution's tails. On the other hand, the data for the period exhibits a skewness value of 0.1825 and a kurtosis value of 3.8797. The positive skewness value shows that the distribution of spiking periods is slightly skewed towards longer period values, with a slightly longer tail on the right side of the distribution. This implies that there is a slightly higher frequency of longer intervals between spikes in comparison to shorter intervals. The kurtosis value of 3.8797, which is slightly greater than 3, suggests that the distribution of periods is mildly leptokurtic. This indicates that the distribution exhibits a slightly more concentrated peak and heavier tails in comparison to a normal distribution. The mildly elevated kurtosis value indicates a slight clustering of period values around the mean, accompanied with a small number of outliers in the distribution's tails.

Fig 7 illustrates the spiking behaviour of Chondroitin-proteinoid at a concentration of 1.8 mg/ml. The raster plot (Fig 7a) shows the temporal distribution of spike events over the recording period. The interspike interval distribution (Fig 7b) provides insights into the temporal patterns of neuronal firing. Analysis of the spike train revealed an average firing rate of 0.004776 spikes/s, indicating relatively sparse spiking activity. The low coefficient of variation (CV) of ISIs (0.22) suggests a fairly regular firing pattern, while the low Fano factor (FF) of spike counts (0.05) indicates minimal variability in the number of spikes across time windows [36]. The Fano factor (FF) is a measure of spike count variability that compares the variance of the spike count to its mean over a specified time window. It is defined as:

$$FF = \frac{\text{Var}(N)}{\text{E}[N]} \tag{7}$$

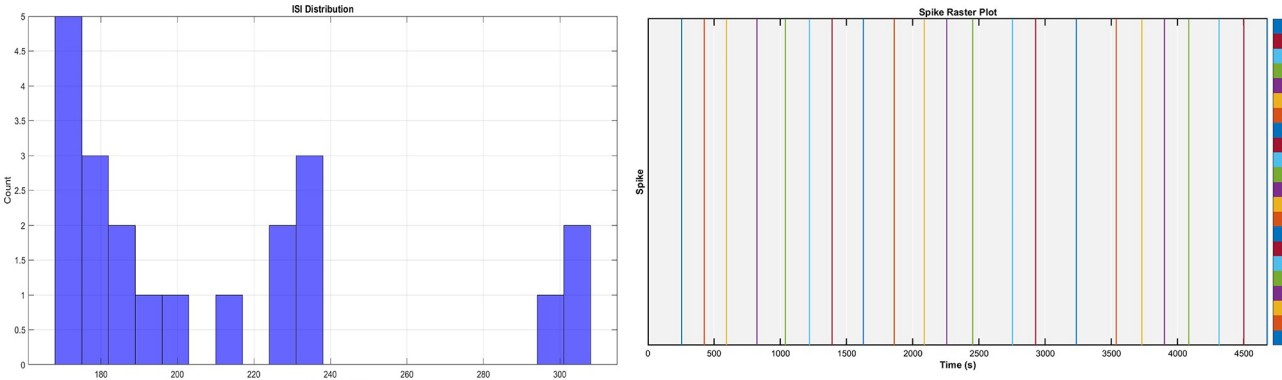

**Fig 7. Spiking dynamics of chondroitin sulfate -proteinoid microspheres at 1.8 mg/ml concentration.** Spiking behavior of Chondroitin Sulfate-proteinoid (1.8 mg/ml). (a) Raster plot showing spike times over the recording period. Each vertical line represents a single spike event. (b) Distribution of interspike intervals (ISIs) in seconds. The average firing rate was 0.004776 spikes/s, with a coefficient of variation (CV) of ISIs of 0.22 and a Fano factor (FF) of spike counts of 0.05.

where Var($N$) is the variance of the spike count and E[$N$] is the expected value (mean) of the spike count. To calculate the FF, the recording time is divided into equal-sized time windows, and the number of spikes in each window is counted. The ratio of the variance to the mean of these spike counts gives the FF. A Fano factor of 1 indicates Poisson-like variability, while values less than 1 suggest sub-Poisson variability (more regular spiking patterns) and values greater than 1 indicate super-Poisson variability (more irregular or bursty patterns). In our analysis of chondroitin-proteinoid microspheres at 1.8 mg/ml, we observed a Fano factor of 0.05, indicating highly regular spiking behaviour with minimal variability in spike counts across time windows.

Fig 8a shows the FF as a function of the observation window size $w$, ranging from 0.1 to 100 seconds. The FF is defined as:

$$FF(w) = \frac{\text{Var}(N(w))}{\text{E}[N(w)]} \tag{8}$$

where $N(w)$ is the spike count in a window of size $w$, Var denotes variance, and E denotes expected value [37]. The consistently low FF values (blue line) relative to the Poisson reference (FF = 1, red dashed line) indicate highly regular spiking patterns with sub-Poisson variability. Fig 8b extends this analysis by comparing the experimental data with theoretical models across different timescales. The x-axis represents both the rate parameter $\lambda$ for theoretical models and $1/w$ for experimental data, allowing direct comparison. The theoretical models include Gamma and Inverse Gaussian renewal processes [38], as well as Markov Poisson Processes (MPP) and Alternating Poisson Processes (APP) [39]. For renewal processes, the FF can be expressed as:

$$FF(w) = 1 - \frac{1 - \int_0^w (1 - F(t))dt}{\lambda w} \tag{9}$$

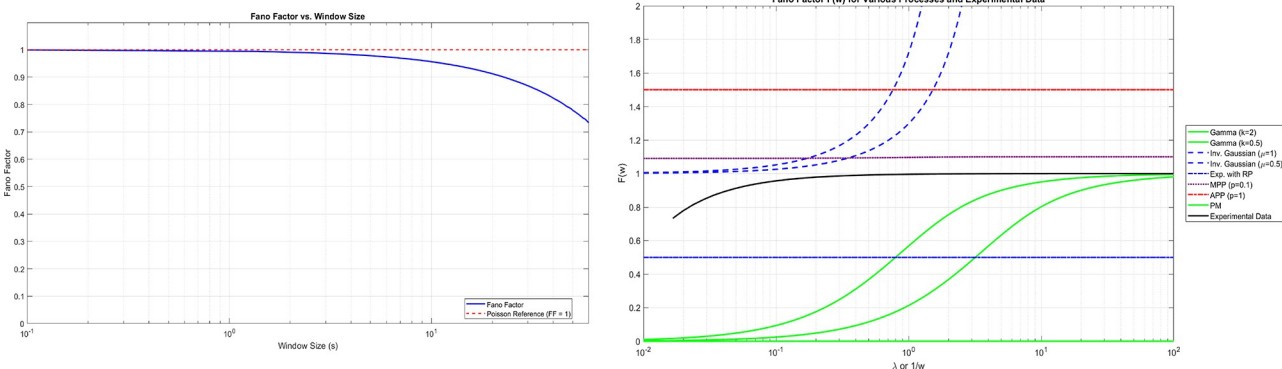

**Fig 8. Comparison of Fano factor F(w) for various theoretical spike train models and experimental data from chondroitine-proteinoid microspheres (1.8 mg/ml).** a) The graph shows the Fano Factor (FF) as a function of the observation window size, ranging from 0.1 to 100 seconds. The blue line represents the calculated FF values, while the red dashed line indicates the reference level for Poisson-like behaviour (FF = 1). The consistently low FF values across different timescales suggest highly regular spiking patterns, with sub-Poisson variability in spike counts. This regularity is maintained even as the observation window increases, indicating stable and structured firing behaviour of the microspheres. b) The graph shows the Fano Factor F(w) as a function of $\lambda$ (rate parameter) or $1/w$ (inverse of window size) on a logarithmic scale. Theoretical models are represented as follows: Green solid lines show Gamma distribution processes with shape parameters k = 2 and k = 0.5. Blue dashed lines represent Inverse Gaussian distribution processes with mean parameters $\mu$ = 1 and $\mu$ = 0.5. The blue dash-dotted line indicates an Exponential distribution with refractory period (RP), F = 0.5. A purple dotted line represents the Markov Poisson Process (MPP) with p = 0.1. The red dash-dotted line shows the Alternating Poisson Process (APP) with p = 1, F = 1.5. A green solid line at F = 0 represents the Periodic Motion (PM) process. The black solid line represents the experimental data from chondroitine-proteinoid microspheres, showing the calculated Fano Factor across different timescales. The x-axis represents both $\lambda$ for theoretical models and $1/w$ (inverse of window size in seconds) for experimental data, allowing direct comparison. The y-axis shows the Fano Factor F(w), indicating the variability in spike counts relative to a Poisson process (F = 1).

where $F(t)$ is the cumulative distribution function of the interspike intervals and $\lambda$ is the mean firing rate [40]. The experimental data (black solid line) exhibits a unique FF profile that deviates from classical spiking models. At shorter timescales, the FF is notably lower than 1, suggesting a more regular firing pattern than a Poisson process. As the timescale increases, the FF shows a gradual rise, indicating increasing variability at longer observation windows. This behaviour contrasts with both the constant FF of a Poisson process (F = 1) (Fig 8a) and the asymptotically approaching FF of renewal processes. It suggests that the chondroitine-proteinoid microspheres exhibit complex temporal dynamics that cannot be fully captured by simple renewal or Markov models. The observed pattern may indicate the presence of multiple timescales in the underlying spiking mechanism, possibly reflecting the complex biochemical processes within the microspheres.

## Izhikevich-based modeling of thalamocortical signal propagation in proteinoid-chondroitin complexes

In this section, we investigated the signal processing capabilities of proteinoid-chondroitin mixtures (1.8 mg/ml) using a thalamocortical network model based on Izhikevich neurons. We used theoretical modeling. It links our findings on electrical activity in proteinoid-chondroitin sulfate complexes to their potential neural-like signaling functions. Our experiments show, especially in the spiking behaviour, that these complexes have dynamics like neural signalling. This resemblance led us to investigate if these systems could mimic complex neural network behaviours. We chose a thalamocortical network model for our analysis for several reasons. The thalamocortical circuit is key to mammalian brains. It helps with sensory processing, attention, and awareness. Secondly, this circuit has a mix of dynamics. It has oscillatory patterns and complex signal transmission. These match the varied spiking behaviour in our proteinoid-chondroitin sulphate complexes. The Izhikevich neurone model is efficient and can replicate various firing patterns. It is the best choice for simulating these dynamics while keeping biological plausibility. Fig 9 presents a comprehensive analysis of the thalamocortical potentials of input and output observed in our experimental setup.

## Izhikevich neuron model

The Izhikevich neuron model, which we used to simulate thalamocortical dynamics, is described by the following system of ordinary differential equations:

$$\frac{dv}{dt} = 0.04v^2 + 5v + 140 - u + I \tag{10}$$

$$\frac{du}{dt} = a(bv - u) \tag{11}$$

with the auxiliary after-spike resetting:

$$\text{if } v \geq 30 \text{ mV, then } \begin{cases} v \leftarrow c \\ u \leftarrow u + d \end{cases} \tag{12}$$

Here, $v$ represents the membrane potential, $u$ is a recovery variable, and $I$ is the input current. The parameters $a$, $b$, $c$, and $d$ are dimensionless parameters that can be adjusted to reproduce various firing patterns observed in cortical neurons.

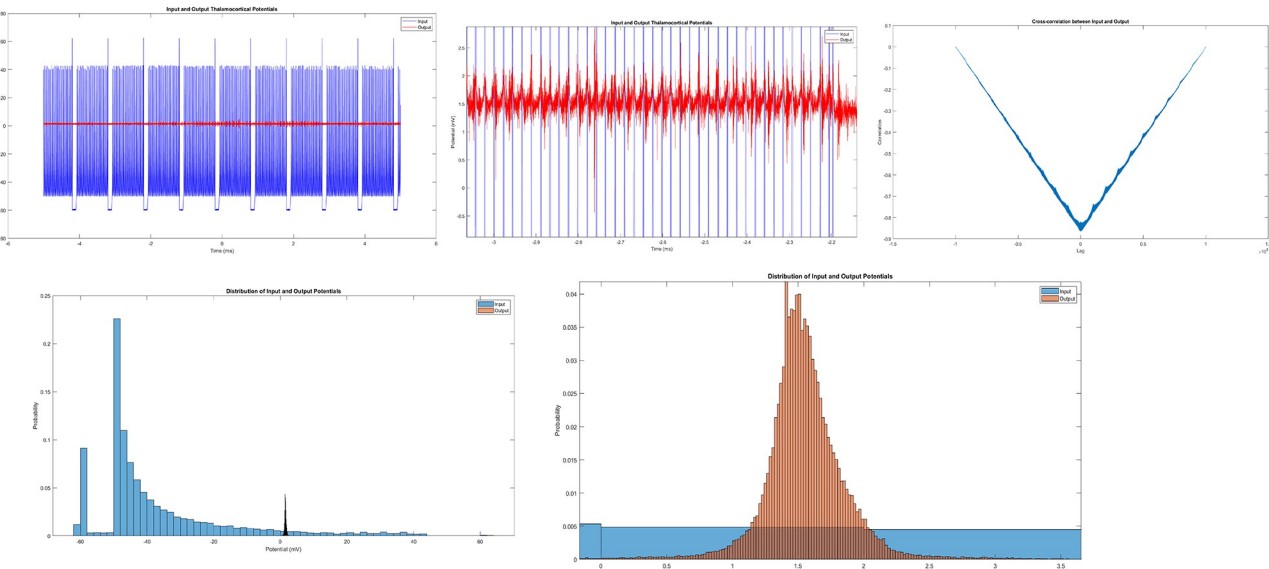

**Fig 9. Analysis of thalamocortical potentials in proteinoid-chondroitin mixture (1.8 mg/ml).** (a) Time series of input (blue) and output (red) thalamocortical potentials. The input shows larger amplitude fluctuations (−60.295 to 62.462 mV) compared to the output (−1.8844 to 4.8428 mV), suggesting signal attenuation or modulation by the proteinoid-chondroitin network. (b) Enlargement of (a) to observe the spiking dynamics of protein-chondroitin complexes. (c) Cross-correlation between input and output potentials. The maximum correlation occurs at a lag of −122 ms, indicating that changes in the output tend to precede similar patterns in the input by this time interval. (d) Probability distribution of input (blue) and output (red) potentials. The input distribution is broader and more symmetric, while the output shows a narrower, slightly right-skewed distribution, reflecting the network's filtering effect. Descriptive statistics. Input potential: mean = −37.413 mV, SD = 20.55 mV; Output potential: mean = 1.5487 mV, SD = 0.32952 mV. The significant difference in means (t = −599.4801, p < 0.0001) suggests substantial signal transformation by the network. (e) Enlargement of the probability distribution of chondroitin-proteinoid complexes.

## Analysis of simulated thalamocortical potentials

As shown in Fig 9a and 9b, the time series of input and output thalamocortical potentials reveal significant differences in signal characteristics. The input potential (blue) exhibits larger amplitude fluctuations (range: −60.295 to 62.462 mV) compared to the output potential (red) (range: −1.8844 to 4.8428 mV). This substantial attenuation and transformation of the signal suggest that the proteinoid-chondroitin network acts as a complex filter, potentially mimicking aspects of biological neural signal processing. The cross-correlation analysis (Fig 9c) reveals a maximum correlation at a lag of −122 ms. This negative lag indicates that changes in the output tend to precede similar patterns in the input, suggesting a predictive or anticipatory behaviour in the network. The probability distributions of input and output potentials (Fig 9d) further illustrate the transformative effect of the proteinoid-chondroitin network. The input distribution is broader and more symmetric, while the output shows a narrower, slightly right-skewed distribution. This change in distribution shape reflects the network's nonlinear filtering properties. Statistical analysis (Fig 9e) quantifies these observations. The input potential has a mean of −37.413 mV (SD = 20.55 mV), while the output potential has a mean of 1.5487 mV (SD = 0.32952 mV). The significant difference in means (t = −599.4801, p < 0.0001) confirms the substantial signal transformation by the network. The correlation analysis (Fig 9e) shows a moderate positive relationship between input and output (Pearson correlation coefficient = 0.3834). The Kolmogorov-Smirnov test (p < 0.00000001) further confirms that the input and output distributions are significantly different. Thalamocortical analysis is also presented in Table 2. Together, these results demonstrate that the proteinoid-chondroitin mixture significantly alters the characteristics of the thalamocortical signal. The observed

**Table 2. Summary of thalamocortical potential analysis in proteinoid-chondroitin mixture (1.8 mg/ml).** This table presents key statistical measures and test results comparing input and output thalamocortical potentials in the proteinoid-chondroitin network. The substantial differences in mean and standard deviation between input and output potentials (input: −37.413 ± 20.55 mV; output: 1.5487 ± 0.32952 mV) indicate significant signal transformation by the network. The highly significant t-test result (t = −599.4801, p < 0.0001) confirms this difference. The moderate positive correlation (r = 0.3834) suggests some preservation of signal features despite the transformation. The Kolmogorov-Smirnov test (p < 0.00000001) further supports the distinct nature of input and output distributions. Notably, the cross-correlation analysis reveals a lag of −122 ms at maximum correlation, indicating that output changes precede similar input patterns. These results collectively demonstrate the complex signal processing capabilities of the proteinoid-chondroitin mixture, potentially mimicking aspects of biological neural networks in thalamocortical signal modulation.

| Measure | Input Potential | Output Potential |
|---|---|---|
| Mean (mV) | −37.413 | 1.5487 |
| Standard Deviation (mV) | 20.55 | 0.32952 |
| Minimum (mV) | −60.295 | −1.8844 |
| Maximum (mV) | 62.462 | 4.8428 |
| **Statistical Tests** | | |
| t-statistic | −599.4801 | |
| p-value (t-test) | < 0.0001 | |
| Correlation Coefficient | 0.3834 | |
| K-S Test p-value | < 0.00000001 | |
| **Cross-correlation** | | |
| Lag at Max Correlation (ms) | −122 | |

transformations in signal amplitude, timing, and distribution suggest that this biomimetic system may capture some key features of biological neural signal processing, potentially offering insights into the role of extracellular matrix components in shaping neural dynamics.

## Discussion

The complex interaction between clusters of CS and proteinoid microspheres [41] might offer insights in mechanisms of proto-synaptic learning mechanisms [8, 42]. The interaction depicted in Fig 10 serves as the basis for a sophisticated and adaptable system that is capable of processing and storing information.

The process of interaction can be defined analytically in the following manner:

Our multiparametric analysis of proteinoid microspheres revealed complex structural and functional characteristics (Fig 11). Edge detection techniques highlighted distinct structural boundaries, providing insight into the microspheres' surface morphology and potential interaction sites (Fig 11A). The inverted and sharpened imaging, simulating high-contrast penetrative analysis, emphasized internal density variations, suggesting heterogeneous composition within individual microspheres (Fig 11B). Furthermore, contrast enhancement combined with Gaussian blur and subsequent sharpening amplified regions of potential molecular excitation or energy absorption, indicating possible reactive sites or areas of functional significance (Fig 11C). The detailed visualization of electrical excitation in proteinoid microspheres (Fig 12) explained key aspects of their electrophysiological action. The stable pre-excitation state of proteinoid microspheres exhibited a compact structure, while the excited state demonstrated notable conformational changes in response to electrical stimulation. Internal excitation, represented by circular arrows, suggests localized molecular rearrangements or charge redistributions within the microsphere matrix. The simplified protein structure illustration emphasizes the fundamental building blocks underlying these complex behaviours.

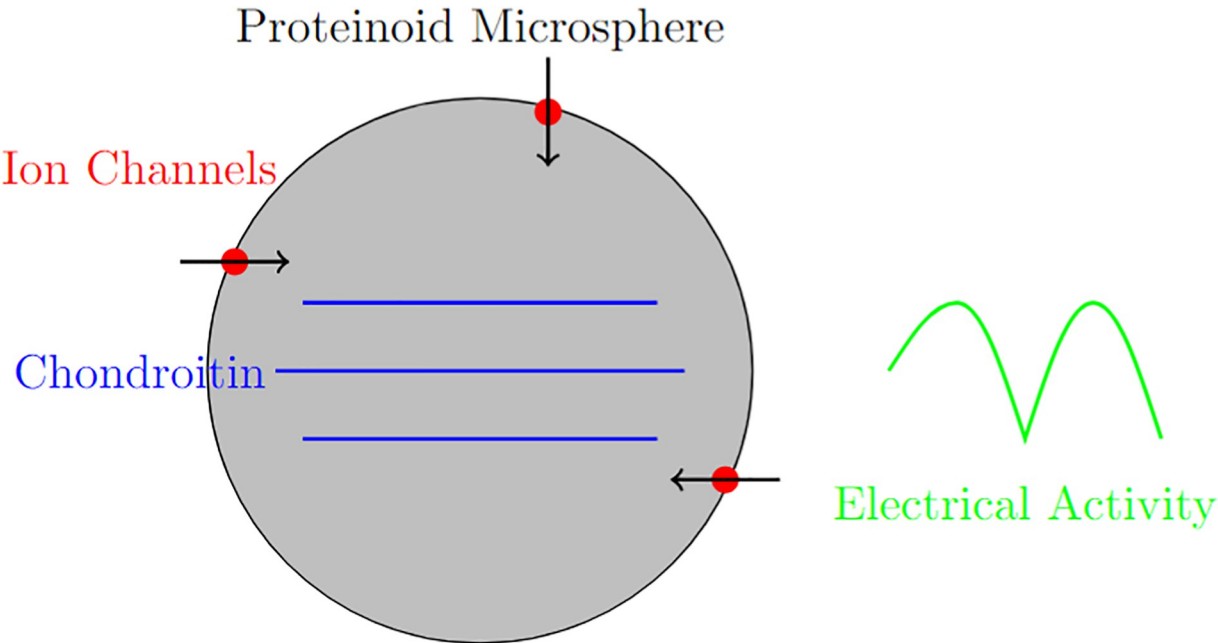

**Fig 10. Mechanism of chondroitin-proteinoid interaction and its computational implications.** The figure illustrates the hypothesized interaction between chondroitin molecules and a proteinoid microsphere. Chondroitin (blue) interacts with the surface of the proteinoid microsphere, potentially modulating ion channels (red). This interaction leads to changes in ion flow (arrows) and subsequent electrical activity (green), which forms the basis for information processing in this system. The unique properties of this interaction can enable unconventional computing paradigms, including distributed information processing, potential memory storage, and adaptive behaviour in response to environmental stimuli.

Quantitative analysis of the microsphere responses to electrical fields, as depicted by the directional arrows in Fig 12, indicates a non-uniform excitation pattern. This heterogeneity in response could be attributed to variations in proteinoid density, charge distribution, or the presence of specific reactive sites within the microsphere structure. The scale bar provided (1 $\mu$m) establishes the nanoscale nature of these phenomena.

### Surface interaction and ion channel modulation

Chondroitin sulphate, a highly charged glycosaminoglycan, forms interactions with the surface of proteinoid microspheres via electrostatic and hydrogen bonding. The mathematical representation for this interaction can be described using a modified Poisson-Boltzmann equation:

$$\nabla \cdot [\epsilon(r)\nabla\psi(r)] = -4\pi\rho_f(r) - 4\pi\sum_i z_i e c_i^0 \exp(-z_i e\psi(r)/k_B T) \tag{13}$$

where $\psi(r)$ is the electrostatic potential, $\epsilon(r)$ is the dielectric constant, $\rho_f(r)$ is the fixed charge

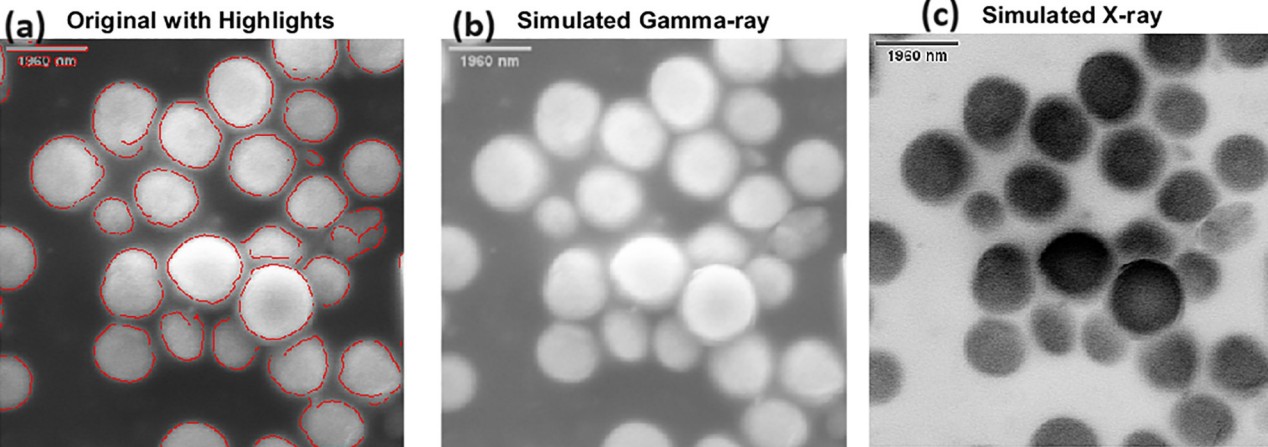

**Fig 11. Multiparametric analysis of proteinoid microspheres.** (A) Original micrograph with edge detection overlay, highlighting structural boundaries. (B) Inverted and sharpened image simulating high-contrast penetrative imaging, emphasizing internal density variations. (C) Contrast-enhanced image with applied Gaussian blur and subsequent sharpening, highlighting potential areas of molecular excitation or energy absorption. These processing approaches allow for the identification of certain structural characteristics and possible functional properties of the microspheres, making it easier to conduct a thorough examination of their arrangement, composition, and physical properties.

density of the CS clusters, $z_i$ is the valence of ion species $i$, $c_i^0$ is the bulk concentration of ion species $i$, $e$ is the elementary charge, $k_B$ is the Boltzmann constant, and $T$ is the temperature.

The behaviour of ion channels embedded in the proteinoid microsphere membrane is affected by this surface interaction [43]. A Boltzmann distribution can model the possibility of an ion channel being in an open state:

$$P_{open} = \frac{1}{1 + \exp(\Delta G / k_B T)} \tag{14}$$

where $\Delta G$ is the difference in the Gibbs free energy between the open and closed states, which is influenced by the CS-proteinoid interaction [44, 45].

## Electrical activity and information processing

Changes in the electrical properties of the microsphere are a result of the modulation of ion channel activity [46]. The Hodgkin-Huxley model incorporates the membrane potential $V_m$:

$$C_m \frac{dV_m}{dt} = -\sum_i g_i (V_m - E_i) + I_{ext} \tag{15}$$

where $C_m$ is the membrane capacitance, $g_i$ are the ion-specific conductances (modulated by the CS interaction), $E_i$ are the reversal potentials for different species of ions and $I_{ext}$ represent external currents.

This electrical activity forms the basis for information processing in the system. Specific activation patterns can be considered as a form of temporal coding, where information is encoded in the timing of electrical events [47, 48].

## Synaptic plasticity and learning

The interaction between CS and the proteinoid introduces a mechanism for synaptic plasticity [49]. Changes in the conformation of proteinoid microspheres or the concentration of CS can

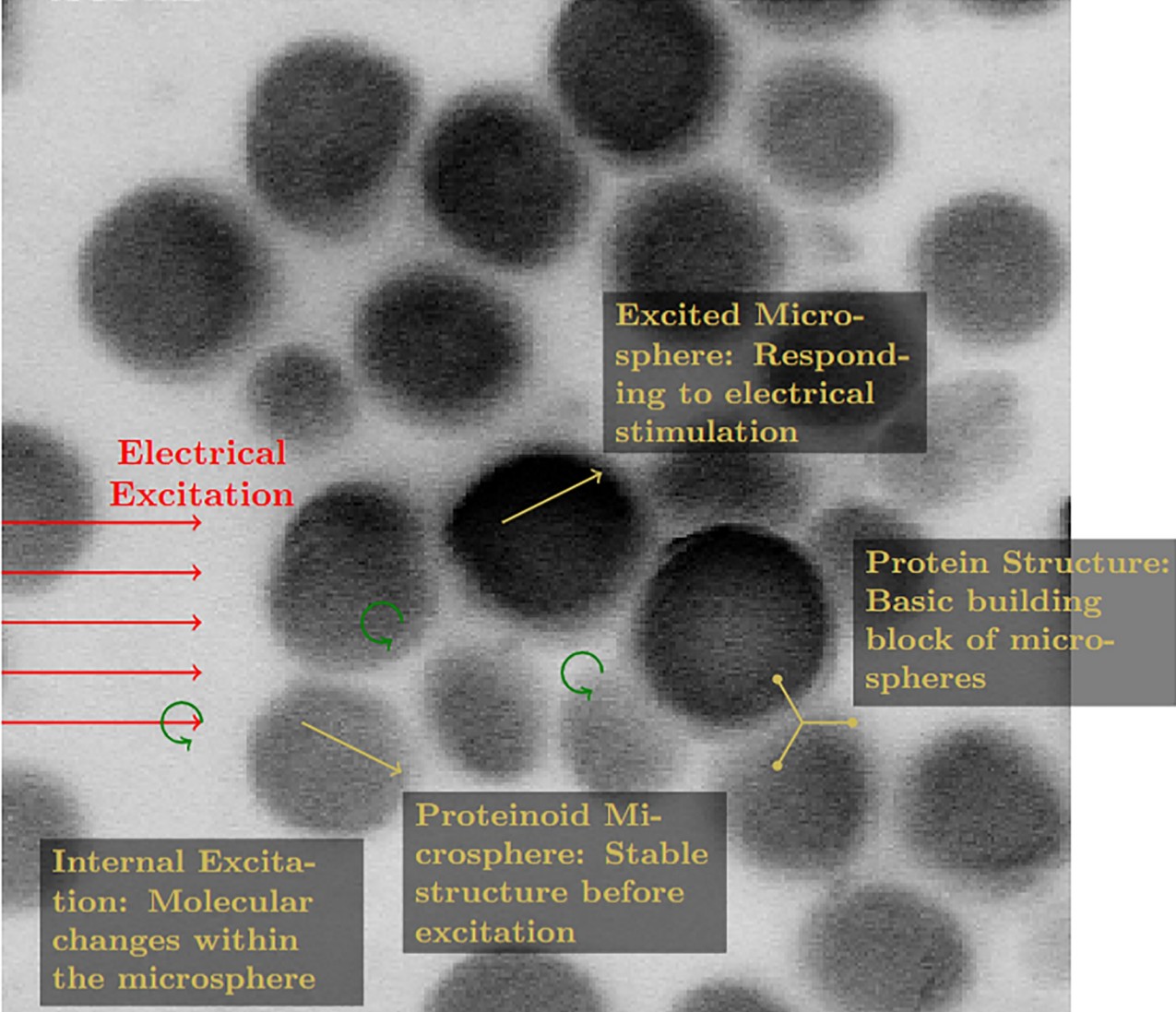

**Fig 12. Detailed visualization of electrical excitation in proteinoid microspheres using simulated X-ray imaging.** The image illustrates the response of microspheres to electrical stimulation, internal excitation processes, and includes a simplified representation of the protein structure.

result in long-term modifications to the electrical properties of the system. We can simulate this plasticity employing a modified Hebbian learning rule [50]:

$$\frac{dw_{ij}}{dt} = \eta(CS_{ij}) \cdot x_i \cdot y_j \tag{16}$$

where $w_{ij}$ represents the strength of the connection between microspheres $i$ and $j$, $\eta(CS_{ij})$ is a

learning rate that depends on the CS concentration, and $x_i$ and $y_j$ represent the activities of microspheres $i$ and $j$ respectively.

## Implications for unconventional computing

As shown in Fig 10, this CS-proteinoid system has several key implications for unconventional computing:

1. *Distributed Information Processing*: The network of interacting microspheres can perform parallel processing of information, potentially enabling complex computations.

2. *Adaptive Memory Storage*: The plasticity of the system allows for the storage of information in the form of altered electrical properties and connectivity patterns.

3. *Environmental Responsiveness*: The system's sensitivity to local chemical environments (e.g., CS concentration) enables adaptive behaviour in response to external stimuli.

Our research on proteinoid microspheres is similar to other self-organising nanoscale systems. This includes the nanoparticle networks studied by Brown and others [51]. Both show emergent, collective behaviours from interactions among individual components. Although the principles vary, they are the same at the core level. The influence of randomness and disorder is probably considerable in our system, although it is not clearly represented in our current methodology. Future research may explore if ideas from disordered systems, informed by studies on memristive networks, could improve our understanding of the dynamics in proteinoid microsphere networks [52, 53]. For instance, examining how network characteristics like as density, average connectedness, and percolation thresholds affect the system's electrical behaviour may provide significant insights [54, 55]. Also, using methods from studies on disordered brain networks [55] could give new ways to understand our system. Though outside this study, these directions offer promise. They may guide research to clarify the principles behind proteinoid microsphere networks and similar self-organizing nanoscale systems. Our results on CS and proteinoid microspheres match recent studies on CS's role in tissue healing and regeneration. Hao et al. found that CS helps tissue repair [56]. It does this by boosting cell growth, movement, and development. It also regulates inflammation and modulates the extracellular matrix. CS-modulated proteinoid microspheres have electrical activity. It may help repair processes, especially in brain tissue. CS can affect the electrical properties of proteinoid structures. This suggests possible uses in neural tissue engineering and regenerative medicine. Future research may explore how CS's electrical regulation affects tissue repair and regeneration, especially in the nervous system. Also, our work improves understanding of how electrical impulses and extracellular matrix components interact to regulate brain activity. Wan et al. studied the interaction between bone and nerves in the skeleton. It emphasized the effect of electrical signals from the bones on nerve growth and function [57]. The electrical activity reported in proteinoid-CS complexes may resemble elements of bone-nerve communication. This analogy suggests our proteinoid-CS system may be a simpler model. It can help examine the complex links among the extracellular matrix, electrical signaling, and central nervous system (CNS) function. Future research may explore how the electrical traits of proteinoid-CS complexes affect neural cells. It could reveal new insights into nerve cell growth and repair. It may also help in treating neurological diseases. In summary, the interaction between proteinoid microspheres and chondroitin sulphate clusters offers a promising model for hybrid neuromorphic systems. The mechanisms described in this article offer a theoretical framework for understanding and potential engineering of such systems for advanced information processing tasks. The experimental validation of these models and the investigation of practical

implementations in fields such as biocomputing and adaptive control systems should be the primary focus of future research.

## Acknowledgments

Authors are grateful to David Paton for helping with SEM imaging and to Neil Phillips for helping with instruments.

## Author Contributions

**Conceptualization:** Andrew Adamatzky.

**Data curation:** Panagiotis Mougkogiannis.

**Formal analysis:** Panagiotis Mougkogiannis.

**Funding acquisition:** Andrew Adamatzky.

**Investigation:** Panagiotis Mougkogiannis.

**Methodology:** Panagiotis Mougkogiannis, Andrew Adamatzky.

**Project administration:** Andrew Adamatzky.

**Resources:** Andrew Adamatzky.

**Supervision:** Andrew Adamatzky.

**Visualization:** Panagiotis Mougkogiannis.

**Writing – original draft:** Panagiotis Mougkogiannis.

**Writing – review & editing:** Andrew Adamatzky.

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
