## [Decision Letter · Decision Letter 0]

11 Oct 2024

PONE-D-24-32280Modulation of electrical activity of proteinoid microspheres with chondroitin sulfate clustersPLOS ONE

Dear Dr. Mougkogiannis,

Thank you for submitting your manuscript to PLOS ONE. After careful consideration, we feel that it has merit but does not fully meet PLOS ONE’s publication criteria as it currently stands. Therefore, we invite you to submit a revised version of the manuscript that addresses the points raised during the review process.

We look forward to receiving your revised manuscript.

Kind regards,

Lei Zhang

Academic Editor

PLOS ONE

2. Thank you for stating the following financial disclosure: [The research was supported by EPSRC Grant EP/W010887/1 ``Computing with proteinoids'']. Please state what role the funders took in the study. If the funders had no role, please state: "The funders had no role in study design, data collection and analysis, decision to publish, or preparation of the manuscript." If this statement is not correct you must amend it as needed. Please include this amended Role of Funder statement in your cover letter; we will change the online submission form on your behalf.

Additional Editor Comments (if provided):

Reviewers' comments:

Reviewer's Responses to Questions

**Comments to the Author**

1. Is the manuscript technically sound, and do the data support the conclusions?

Reviewer #1: Yes

Reviewer #2: Yes

2. Has the statistical analysis been performed appropriately and rigorously? 

Reviewer #1: Yes

Reviewer #2: Yes

3. Have the authors made all data underlying the findings in their manuscript fully available?

Reviewer #1: Yes

Reviewer #2: Yes

4. Is the manuscript presented in an intelligible fashion and written in standard English?

Reviewer #1: Yes

Reviewer #2: Yes

5. Review Comments to the Author

Reviewer #1: This paper investigates the modulation of the electrical activity of proteinoid microspheres via interaction with chondroitin sulfate (CS) clusters. The authors aim to explore how this hybrid system, combining proteinoids—synthetic, neuron-like structures—with components of the brain’s extracellular matrix (CS), could mimic aspects of synaptic plasticity and learning. The findings suggest that CS clusters significantly influence the electrical behavior of proteinoid microspheres, especially in terms of spiking dynamics, offering new insights into molecular mechanisms underlying learning and information processing. The study stands out as an innovative attempt to combine biomimetic structures with complex biochemical networks, with potential applications in neuromorphic computing and unconventional bio-inspired systems.

What makes this work novel is its attempt to merge synthetic proteinoids with natural biomolecules like CS, creating a hybrid system that could provide clues about the origins of neuron-like communication. This reminds me of research on memristive nanoparticles, where complex interactions at nanoscale interfaces lead to adaptive electrical behavior, suggesting parallels in the system’s potential to store and process information.

Given the adaptive and spike-driven nature of this hybrid system, have you considered exploring its suitability for reservoir computing? The spontaneous and externally modulated spiking behavior seems to be a promising candidate for such computational architectures.

Since the model they use for this type of analysis is essentially an integrate and fire model, I wonder if they could say anything about reservoir computing with this type of systems and how the random network can affect the information processing.

As far as I know, they only work that does something like this is this one: https://arxiv.org/abs/2407.20547, or this https://www.nature.com/articles/s44335-024-00002-4 although they do not call it RC.

I also have questions about this type of nanoparticles. The system reminds me of the type of nanoparticles studied by Simon Brown. The system is different. See for instance: https://www.science.org/doi/10.1126/sciadv.aaw8438. I wonder whether randomness will play a role. There are not many studies about this, except for those with memristive systems. I don't think the model they use is memristive, but maybe some parallels could be drawn. Analytically, it would be hard to do anything without using techniques from disordered systems (for memristive devices, which I know a bit better, examples are these: https://onlinelibrary.wiley.com/doi/10.1002/andp.202300090, https://www.science.org/doi/10.1126/sciadv.abh1542), but for SNN maybe the standard papers are those by John Hertz and collaborator. Of course, I am not asking to do these calculations, but directions that theorists and experimentalists can investigate along the line of density, percolation transitions, average connectivity (Etc). See for instance https://arxiv.org/pdf/1606.04466 or https://arxiv.org/html/2409.00412v1.

I think this paper is very nice, I enjoyed reading it quite a lot.

Minor corrections:

- Clarify the dosage impact on electrical activity, particularly how variability in electrical potential changes across CS concentrations.

- Add more details on the statistical methods used for analyzing spike data.

- The transition between experimental results and theoretical modeling could benefit from more explanatory context.

Reviewer #2: Mougkogiannis et al. worked on the interaction between proteinoid microspheres and chondroitin sulfate (CS) clusters, as well as how this interaction regulates the electrical activity of the proteinoid microspheres. The findings hold significance for understanding the application of biomaterials in neural system simulation, especially regarding the comprehension of molecular mechanisms underlying synaptic plasticity. The authors have a clear idea throughout the article. The pictures are very intuitive and interesting. The manuscript may be further improved by considering following points:

1. Introduction:

Previous studies have reported that proteloid microspheres can exhibit electrical spikes. Therefore, it could more clearly highlight the innovative aspects of this study, such as the experimental methods, theoretical models, or new mechanisms expected to be discovered.

2. Results:

- There are discrepancies between the numbering of some figures and their references within the manuscript. For example, Figure 2 is mentioned but does not appear to correspond to the described content

-The abbreviation of chondroitin sulfate has been written as CS, so do not write the full name in the following text.

3. Discussion:

-The discussion section still reads more like a restatement of the research findings rather than a comprehensive summary and reflection on the content of the study. Moreover, the limitations of the current research have not been adequately addressed. It would be beneficial to expand on these points.

-Appropriate relevant literature can be cited to discuss the potential applications in a broader biological or medical context. A recent article about CS promoting tissue repair should provide the needed references https://doi.org/10.1016/j.carbpol.2023.120738. I also recommend the authors read and cite the following review about how electrical signals synergize with matrix cells to regulate neural functions: https://doi.org/ 10.1002/advs.202003390.

4. Reference:

Please ensure the reference format complies with the journal's requirements.

6. PLOS authors have the option to publish the peer review history of their article (what does this mean?). If published, this will include your full peer review and any attached files.

Reviewer #1: No

Reviewer #2: No

---

## [Author Response · Author response to Decision Letter 0]

16 Oct 2024

Please find the attached file with the response to the reviewers.

---

## [Editor Report · Decision Letter 1]

18 Oct 2024

Modulation of electrical activity of proteinoid microspheres with chondroitin sulfate clusters

PONE-D-24-32280R1

Dear Dr. Mougkogiannis,

We’re pleased to inform you that your manuscript has been judged scientifically suitable for publication and will be formally accepted for publication once it meets all outstanding technical requirements.

Kind regards,

Lei Zhang

Academic Editor

PLOS ONE
---

## [Editor Report · Acceptance letter]

23 Oct 2024

PONE-D-24-32280R1 

PLOS ONE

Dear Dr. Mougkogiannis, 

I'm pleased to inform you that your manuscript has been deemed suitable for publication in PLOS ONE. Congratulations! Your manuscript is now being handed over to our production team.

Kind regards, 

on behalf of

Dr. Lei Zhang 

Academic Editor

PLOS ONE